# Exploring the Hidden Dimension in Accelerating Convolutional Neural Networks

## Abstract

DeePa is a deep learning framework that explores parallelism in all parallelizable dimensions to accelerate the training process of convolutional neural networks. DeePa optimizes parallelism at the granularity of each individual layer in the network. We present an elimination-based algorithm that finds an optimal parallelism configuration for every layer. Our evaluation shows that DeePa achieves up to $6.5\times$ speedup compared to state-of-the-art deep learning frameworks and reduces data transfers by up to $23\times$.

## 1 Introduction

Training convolutional neural networks (CNNs) is increasingly compute-intensive and time-consuming. It takes days or even weeks to train deep CNNs from scratch (Szegedy et al., 2014; Zeiler & Fergus, 2014; Simonyan & Zisserman, 2014; Szegedy et al., 2016). Existing deep learning frameworks such as TensorFlow, PyTorch, and Caffe2 parallelize the training process onto multiple processors (usually GPUs) using *image parallelism*[1] dividing the entire image dataset into batches with the same number of images and assigning each batch to a dedicated processor.

The standard parallelization of CNN training only exploits image parallelism. However, other dimensions can also parallelize the training process. For example, in CNNs for 2D images, data is commonly organized as 4-dimensional tensors (i.e., image, height, width, channel). The *image* dimension includes an index for each image in the input dataset. The *height* and *width* dimensions specify a position in an image. For a particular position, the *channel* dimension[2] indexes different neurons for that position. Exploring these other parallelizable dimensions can potentially reduce the compute time and data transfer cost when training CNNs (see Section 2). Moreover, different layers in a CNN may prefer different parallelism configurations for achieving optimal performance.

We propose DeePa, a deep learning framework that explores parallelism in all parallelizable dimensions to accelerate the training of CNNs. To the best of our knowledge, DeePa is the first system that models and exploits the parallelism of neural networks at the granularity of each individual layer. To generate a parallelism configuration for each layer, DeePa uses an elimination-based algorithm that automatically finds the configuration with the best estimated performance.

The main contributions of this paper are:

- We present DeePa, a deep learning framework that explores parallelism in all parallelizable dimensions to accelerate the training of CNNs.

- The parallelization strategy is selected at the granularity of each individual layer.

- We present an elimination-based algorithm for finding the parallelism configuration with optimal estimated performance for each layer.

- Our evaluation shows that, compared to state-of-the-art deep learning frameworks (e.g., TensorFlow and PyTorch), DeePa achieves $6.5\times$, $1.9\times$, and $1.5\times$ speedup for AlexNet,

---

[1]Some papers use the term *data parallelism* to refer to parallelism across images. Since this paper involves parallelizing the training dataset in other data dimensions, we use *image parallelism* to distinguish this from other parallelization strategies.

[2]Some papers use the term *depth* to refer to different neurons for a position. In this paper, depth refers to the number of layers for an entire neural network and we use *channel* for the neurons for a position.

Table 1: Detailed information for the example convolutional layers used in Figure 1 and 2.

| Name | Input Channels | Output Channels | Height | Width | Kernel | Stride | Description |
|---|---|---|---|---|---|---|---|
| C1 | 128 | 128 | 112 | 112 | 3x3 | 1x1 | Conv4 in VGG-16 |
| C2 | 512 | 512 | 28 | 28 | 3x3 | 1x1 | Conv8, Conv9, and Conv10 in VGG-16 |
| C3 | 192 | 64 | 35 | 35 | 1x1 | 1x1 | Conv1x1 in an Inception-v3 module |
| C4 | 48 | 64 | 35 | 35 | 5x5 | 1x1 | Conv5x5 in an Inception-v3 module |
| C5 | 64 | 192 | 27 | 27 | 5x5 | 1x1 | Conv2 in AlexNet |
| C6 | 256 | 256 | 13 | 13 | 3x3 | 1x1 | Conv5 in AlexNet |
| C7 | 32 | 64 | 147 | 147 | 3x3 | 1x1 | Conv3 in Inception-v3 |
| C8 | 448 | 384 | 8 | 8 | 3x3 | 1x1 | Conv3x3 in an Inception-v3 module |

VGG-16, and Inception-v3, respectively. The performance improvement comes from reducing overall data transfers, automatically overlapping computation with data movement, and accelerating computation throughput.

## 2 MOTIVATION

This work is motivated by the following observations.

### 2.1 ACCELERATING COMPUTATION THROUGHPUT

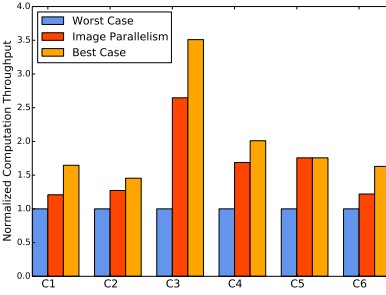

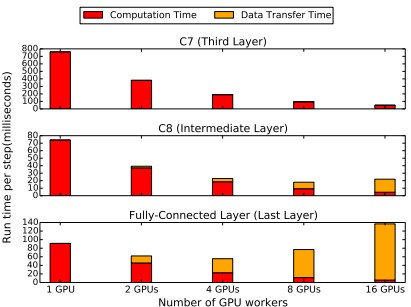

Figure 1: Relative performance for training different convolutional layers. Computation throughput is calculated by dividing the batch size with computation time (both forward processing and back propagation) and is normalized by the worst case.

Figure 2: Computation and data transfer time to process a batch of 512 images using image parallelism for the third layer, an intermediate layer, and the last layer of Inception-v3.

Convolutional layers generally consume the bulk of the training time in CNNs, and parallelizing training in different data dimensions results in significantly different performance. Figure 1 shows the relative speed of training six different convolutional layers from AlexNet, VGG-16, and Inception-v3. The properties of the convolutional layers are shown in Table 1. For each convolutional layer, we tried parallelizing the computation in each individual parallelizable dimension as well as combinations of different parallelizable dimensions, and we report the performance of the standard parallelization over images along with the worst and best parallelization strategies we discovered. Figure 1 shows that different parallelism configurations result in very different performance, and image parallelism generally achieves suboptimal performance. Therefore, exploring parallelism in other dimensions can potentially accelerate the training of convolutional layers.

### 2.2 REDUCING DATA TRANSFER COST

Different parallelization strategies can also result in significantly different amounts of data movement. Figure 3 shows an example of parallelizing the first fully-connected layer of VGG-16 on two GPUs in different dimensions. In image parallelism (Figure 3a), each GPU processes a batch of images and computes the gradient for the entire fully-connected layer. This requires each GPU to synchronize the gradients for the entire fully-connected layer (shown as the shadow rectangles) after each step. An alternative approach (Figure 3b) parallelizes in the channel dimension by assigning

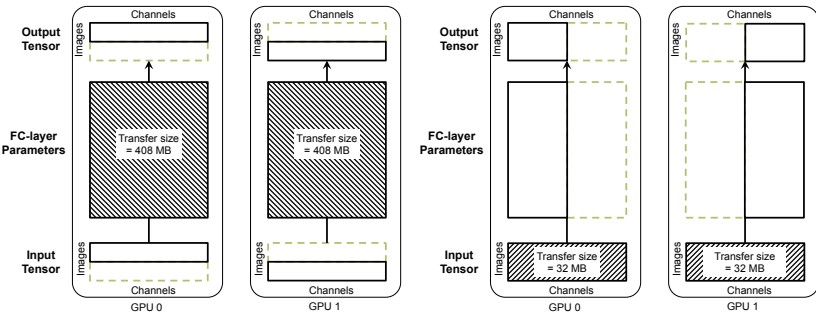

(a) Parallelism in the image dimension.  (b) Parallelism in the channel dimension.

Figure 3: Different configurations for parallelizing the first fully-connected layer of VGG-16. Rectangles with solid lines indicate tensors managed by the local GPU, while rectangles with dot lines are tensors managed by a remote GPU. The shadow rectangles indicate data transfers for each step.

a subset of the output channels to each GPU. As a result, different GPUs compute the gradients for disjoint subsets of the fully-connected layer, which eliminates transferring the fully-connected layer but introduces additional data transfers for input tensors (shown as the shadow rectangles). For this particular case, using parallelism in the channel dimension reduces data transfer costs by $12\times$.

## 2.3 Optimizing per-layer performance

When processing a batch of images, increasing the number of workers does not always improve overall execution time, due to the data transfer overhead to synchronize gradients across different workers. Figure 2 shows the per-step training time for three different layers in Inception-v3 for a batch size of 512 images on up to 16 GPUs. The training time includes forward processing, backward propagation, and gradient aggregation. The figure shows that different layers in a neural network may prefer different hardware configurations, and there is no single configuration that is optimal for all layers. For example, the third layer performs best on 16 GPUs while the last layer performs best on 4 GPUs. Thus, a parallelism configuration includes both selecting the data dimensions to be parallelized and the number of parallel workers (or, equivalently, the number of subsets into which the data is partitioned).

## 3 DeePa

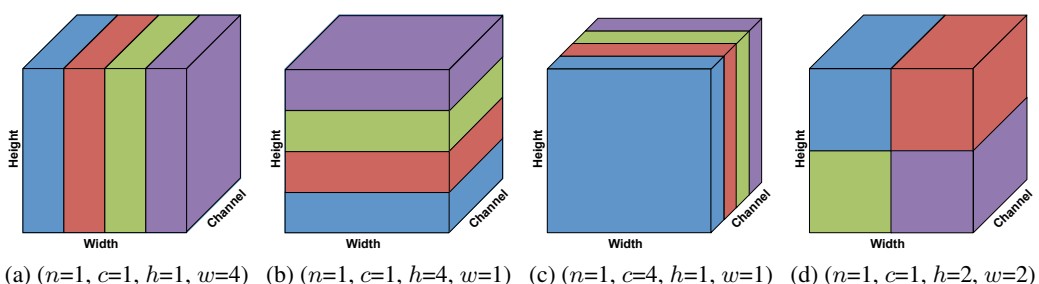

(a) ($n$=1, $c$=1, $h$=1, $w$=4)  (b) ($n$=1, $c$=1, $h$=4, $w$=1)  (c) ($n$=1, $c$=4, $h$=1, $w$=1)  (d) ($n$=1, $c$=1, $h$=2, $w$=2)

Figure 4: Example configurations that parallelize an operation in a single dimension or combinations of multiple dimensions. The figure shows how each image is partitioned in different configurations. Each configuration needs a total of 4 workers.

Similar to TensorFlow and PyTorch, DeePa uses *computation graphs* to describe dependencies between operations. In a computation graph $G = (V, E)$, each node $n \in V$ is an operation (e.g., a convolution or matrix-multiply), and each directed edge $(u, v) \in E$ is a tensor that is an output of $u$ and an input of $v$.

One key difference between DeePa and TensorFlow or PyTorch is that each node in the DeePa computation graph also includes a *configuration* that describes how the corresponding operation is

parallelized across different workers. For each parallelizable dimension (i.e., image, height, width, and channel), the configuration includes an integer that describes the degree of parallelism in that dimension. For a configuration, the product of the integers over all dimensions is the number of workers needed to process the operation in that configuration. Figure 4 demonstrates some example configurations that explore parallelism in a single dimension as well as combinations of different dimensions. DeePa assumes equal partitioning in each dimension. As a result, each worker receives the same size input, which provides well-balanced workload distribution in our experiments.

For each node in the computation graph, its configuration describes how the *output* tensor is divided onto multiple workers. Each worker computes a *disjoint* subset of the output tensor, and thus each worker can process the operation in parallel without data dependencies. Given a node's configuration, DeePa calculates the input sets for each worker and automatically schedules proper data transfers between operations.

DeePa also provides three additional functions:

- For each node $v$ and configuration $c$, $v.compute(c)$ estimates the time to process the corresponding operation under the parallelism configuration $c$. This includes both the forward processing and back propagation time and is estimated by running the operation in that configuration multiple times on the device and measuring the average execution time.

- For each edge $e = (u, v)$, $e.xfer(c_u, c_v)$ estimates the time to transfer the input tensor $e$ to each worker, using the size of the data to be moved and the known communication bandwidth. Note that $e.xfer(c_u, c_v)$ is zero if $u$ and $v$ have the same configuration (i.e., $c_u = c_v$), in which case no data is transferred. As with *compute()*, we precompute the *xfer()* function for each edge in the graph by calculating the overall data transfer size for all possible source and destination configurations.

- For each node $v$ and configuration $c$, $v.update(c)$ estimates the time to update parameters for the corresponding operation. We use the data transfer time to approximate the update time, since the data transfer time is much longer than the compute time for updating parameters. Note that different configurations can have significantly different update time, as described in Section 2.2.

A *global configuration $g$* includes a parallelism configuration for each node in a computation graph: $g(v)$ describes the parallelism configuration for node $v$. Using the functions defined above, we can model the per-step execution time for a computation graph:

$$Cost(g, (V, E)) = \sum_{v \in V} \{v.compute(g(v)) + v.update(g(v))\} + \sum_{e=(u,v) \in E} e.xfer(g(u), g(v)) \quad (1)$$

$Cost(g, (V, E))$ estimates the per-step execution time if the computation graph $(V, E)$ is parallelized using global configuration $g$. This execution time includes forwarding processing, backward propagation, and gradient aggregation. Equation 1 expresses the problem of finding the configuration for each individual node as a global optimization problem.

# 4 FINDING OPTIMAL GLOBAL CONFIGURATIONS

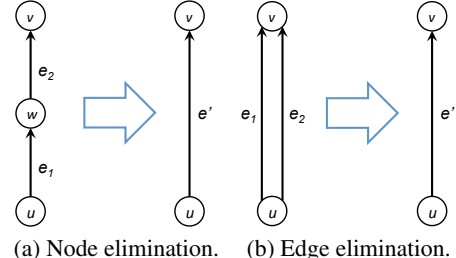

(a) Node elimination.      (b) Edge elimination.

Figure 5: Performing a node/edge elimination on a computation graph.

We now describe our algorithm for finding a global configuration that minimizes Equation 1. In DeePa, each node can select any of a fixed (but large) set of parallelism configurations. Therefore the number of potential global configurations is exponential in the number of nodes in a computation graph, which makes it impractical to enumerate all global configurations for deep CNNs such as VGG-16 and Inception-v3.

However, the CNNs we have seen in practice exhibit strong locality: each node is only connected to a few nodes with similar depth in a computation graph. Based on this observation, we use the following two elimination strategies to iteratively simplify the computation graph while preserving the globally optimal configuration.

**Node elimination**. For each node $w$ with a single in-edge $e_1 = (u, w)$ and a single out-edge $e_2 = (w, v)$, we remove node $w$ and the two edges $e_1$ and $e_2$ from the graph and insert a new edge $e' = (u, v)$ (shown in Figure 5a). The *xfer*() function for node $e'$ is

$$e'.xfer(c_u, c_v) = \min_{c_w}\{e_1.xfer(c_u, c_w) + w.compute(c_w) + w.update(c_w) + e_2.xfer(c_w, c_v)\} \quad (2)$$

Note that because we have precomputed the *xfer*() function for edges in the original graph, we can similarly compute the *xfer*() function for the transitive edge added by a node elimination; i.e., we use dynamic programming to compute the optimal configuration for node $w$ for every possible choice of configurations for nodes $u$ and $v$. For CNNs with a linear computation graph (e.g., AlexNet and VGG-16), node elimination is sufficient to reduce the original graph to a graph with only 2 nodes.

**Edge elimination**. For two edges with the same source and destination node (i.e., $e_1 = (u, v)$ and $e_2 = (u, v)$), we can remove $e_1$ and $e_2$ from the graph and insert a new edge $e' = (u, v)$ (shown in Figure 5b). The *xfer*() function for node $e'$ is

$$e'.xfer(c_u, c_v) = e_1.xfer(c_u, c_v) + e_2.xfer(c_u, c_v) \quad (3)$$

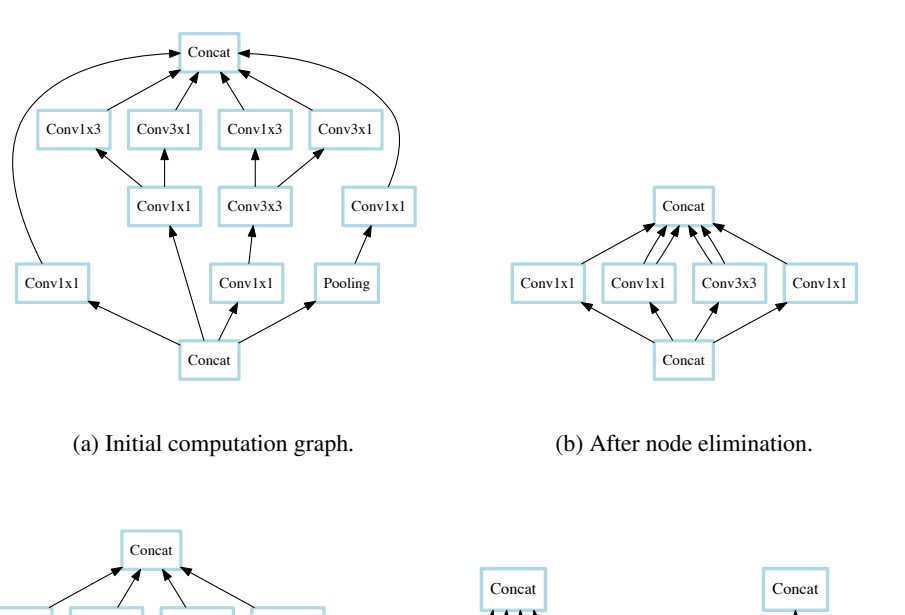

(a) Initial computation graph.

(b) After node elimination.

(c) After edge elimination.

(d) After node elimination.

(e) After edge elimination.

Figure 6: Iteratively performing node/edge eliminations on an Inception module.

As with node elimination, we compute the *xfer*() function for $e'$ using the already computed *xfer*() functions for $e_1$ and $e_2$. Figure 6 shows how DeePa iteratively eliminates nodes and edges for an

Inception-v3 module. The full Inception-v3 computation graph has 120 nodes, which DeePa reduces to a 2-node graph.

DeePa iteratively uses node and edge eliminations to simplify a computation graph until neither elimination can be applied. DeePa then enumerates all global configurations for the final graph and chooses the one that minimizes the *Cost* function in Equation 1.

After deciding the configuration for each node in the final graph, DeePa then decides the configuration for the eliminated nodes by undoing the node and edge eliminations in reverse order. When undoing a node elimination for node $w$, DeePa selects the configuration that minimizes Equation 2 for node $w$. After undoing all eliminations, DeePa has a configuration for every node in the original graph. In Appendix A.1, we prove that our algorithm finds an optimal global configuration. In our experiments, DeePa finds an optimal configuration for parallelizing the largest CNN we have worked with, Inception-v3, on 16 GPUs in about 100ms.

## 5    IMPLEMENTATION

We found that it is non-trivial to parallelize the training of CNNs in the height, width, and channel dimensions in existing frameworks (e.g., TensorFlow, PyTorch, and Caffe2), and none provides an interface for controlling per-operation parallelism. We implemented DeePa in Legion (Bauer et al., 2012), a high-performance parallel runtime for distributed heterogeneous architectures, and use cuDNN (Chetlur et al., 2014) and cuBLAS (cub, 2016) as the underlying libraries for processing neural network operations. The following Legion features significantly simplify our implementation for DeePa. First, Legion supports high-dimensional partitioning that allows us to parallelize any operation in any combination of the dimensions. Second, Legion allows DeePa to control parallelism at the granularity of each operation. Third, Legion allows fine-grain control over the placement of data in memory. Fourth, Legion's asynchronous tasking model makes it easy to exploit task as well as image parallelism. We also include two critical optimizations that help achieve good performance.

**Overlapping computation with data transfers**. DeePa manages the gradients of each operation separately and transfers an operation's gradients as long as its back propagation is completed. We have found that this can effectively hide the data transfer overhead for gradient synchronization. As a result, the synchronous training performance matches asynchronous training in DeePa, which allows users to use synchronous training with its better algorithmic efficiency.

**Distributing parameter servers**. Existing frameworks use *parameter servers* to store and update variables for a CNN model. Parameter servers are located in CPU memory in TensorFlow and PyTorch. Because DeePa manages the parameters for each operation separately, DeePa can opportunistically distribute the parameter server onto the GPU memories whenever possible. This eliminates data transfers for operations whose gradients and parameter server are located on the same GPU and transforms all GPU to CPU copies into faster GPU to GPU copies.

## 6    RELATED WORK

To the best of our knowledge, DeePa is the first deep learning framework that controls and optimizes the parallelism of neural networks in all dimensions at the granularity of each operation.

Existing frameworks such as TensorFlow (Abadi et al., 2016), Caffe2 (Caf, 2016), and PyTorch (Pyt, 2017) use image parallelism to distribute the training of CNNs and only explore parallelism in the image dimension. The standard image parallelism configuration keeps a replica of the entire network on each worker, which results in large data transfers for synchronizing the gradients in each step.

Mirhoseini et al. (2017) uses *model parallelism* that assigns each operation to a dedicated processor for training Inception-v3. It uses a reinforcement learning algorithm to optimize the placement of each operation on a GPU device. The learned device placement on 4 GPUs achieves 19% speedup compared to single GPU performance. However, parallelism in each operation is not explored.

Krizhevsky (2014) introduces "one weird trick" (OWT) that combines image parallelism with model parallelism to accelerate the distributed training of AlexNet, which efficiently reduces the data transfer cost compared to the standard image parallelism configuration. In Section 7.1.2, we show that

DeePa further reduces the overall data transfers for AlexNet by $3\times$ and the per-step training time by $2.3\times$ compared to OWT.

Goyal et al. (2017) empirically shows no loss of accuracy for training ResNet-50 on the ImageNet dataset with a large *minibatch* size of 8192 images[3]. It uses the standard image parallelism configuration to distribute the training onto 256 GPUs and includes a number of optimizations for reducing communication overhead. As communication is a bottleneck in distributed deep learning, we believe our techniques for reducing data transfers can substantially benefit training on large numbers of GPUs.

# 7 EVALUATION

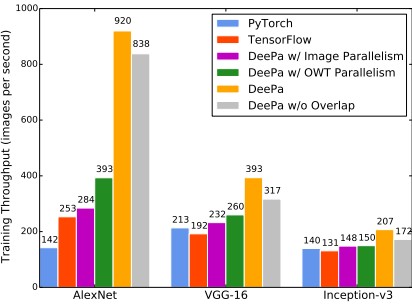

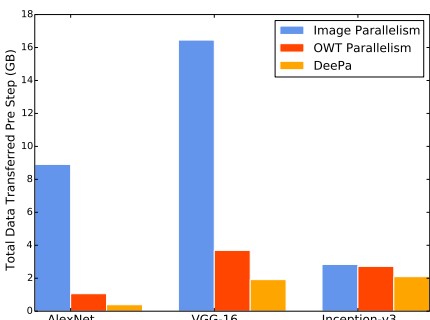

Figure 7: Training throughput (images/second) for AlexNet, VGG-16, and Inception-v3 on 16 GPUs. The purple and green bar shows the DeePa performance by restricting DeePa to use image and OWT parallelism, respectively.

Figure 8: Total amount of data transferred in each step for training AlexNet, VGG-16, and Inception-v3 on 16 GPUs with a minibatch size of 512.

We use AlexNet (Krizhevsky, 2014), VGG-16 (Simonyan & Zisserman, 2014), and Inception-v3 (Szegedy et al., 2016) as benchmark CNNs and use the ImageNet dataset (Russakovsky et al., 2015) as the input. For each CNN, we compare the performance of DeePa against TensorFlow, PyTorch, and OWT. We implement OWT in DeePa by restricting all convolutional and pooling layers to use image parallelism and all fully-connected layers to use model parallelism.

## 7.1 CASE STUDY ON A 16-GPU MACHINE

We conduct a detailed case study for training the three CNNs on a 16-GPU machine, with two Intel 10-core E5-2680 Xeon processors, 256 GB main memory, and 16 NVIDIA Tesla K80 GPUs[4]. We use all 16 GPUs for training each CNN model with a minibatch size of 512 images. As a result, each GPU processes a batch of 32 images in the image parallelism configuration. DeePa uses the search algorithm in Section 4 to find the optimal parallelism configurations, which requires 0.7, 1.1, and 4.8 seconds for AlexNet, VGG-16, and Inception-v3, respectively.

Figure 7 shows the synchronous training throughput for a minibatch size of 512 images on 16 GPUs. When DeePa uses image parallelism for all operations, DeePa achieves competitive performance compared to the best of TensorFlow and PyTorch. The OWT approach that uses model parallelism for fully-connected layers speeds up the training throughput by $1.4\times$, $1.2\times$, and $1.07\times$ compared to image parallelism using DeePa. The best configurations found by DeePa achieve $6.5\times$, $1.9\times$, and $1.5\times$ speedup compared to TensorFlow and PyTorch.

Three main optimizations in DeePa achieve most of the performance benefit over the other frameworks. First, DeePa significantly reduces data transfers in each step, as shown in Figure 8. Compared to image parallelism, the OWT approach reduces data transfers by $1.05\text{-}8.4\times$. However, the best configuration used by DeePa further reduces data transfers by $1.2\text{-}2.7\times$ compared to OWT. Second, the optimization for overlapping computation with data transfers (described in Section 5) effectively hides data transfer latency and achieves better GPU utilization. The grey bars in Figure 7

---

[3]In SGD, the parameters are updated after processing a *minibatch* of training examples.

[4]The machine is equipped with 8 GPU cards, each of which has 2 Tesla K80 GPUs.

Table 2: The cost for different configurations for the first fully-connected of AlexNet.

| Configuration | # GPU Workers | Transfer Cost | Compute Cost | Update Cost | Total Cost | Parallelism Configurations |
|---|---|---|---|---|---|---|
| {n=16, c=1} | 16 | 0 | +1.28 | +1075 | = 1076 | Image Parallelism |
| {n=1, c=16} | 16 | 134.4 | +1.28 | +0 | = 135.7 | OWT |
| {n=1, c=4} | 4 | 33.6 | +5.1 | +0 | = 38.7 | |
| {n=1, c=2} | 2 | 16.8 | +10.2 | +0 | = **27.0** | DeePa |
| {n=1, c=1} | 1 | 8.4 | +20.4 | +0 | = 28.8 | |

illustrate DeePa's performance when the overlap optimization is disabled, which shows that overlapping computation with data transfers can improve the training throughput by 10%-30%. Third, DeePa also improves performance by exploring parallelism in the height and width dimensions (see Section 7.1.3).

### 7.1.1 THE BEST CONFIGURATIONS

We describe the best configurations discovered for AlexNet, VGG-16, and Inception-v3 in Sections 7.1.2 to 7.1.4. The best configurations have several similarities.

First, for the beginning layers with large height/width dimensions and small channel dimensions, DeePa uses image parallelism on all available GPUs, since the data transfers for synchronizing gradients are much smaller than the data transfers for moving tensors between operations.

Second, deeper layers in CNNs tend to have smaller height/width dimensions and larger channel dimensions. As a result, the cost for moving tensors between different operations decreases, while the cost for synchronizing gradients increases. DeePa adaptively reduces the number of GPU workers for these layers to reduce the expensive data transfers for synchronizing gradients at the cost of introducing cheaper data transfers for moving tensors.

Third, DeePa uses model parallelism on a small number of GPU workers for fully-connected layers, because synchronizing gradients and moving tensors are both much more expensive than the compute time for fully-connected layers. DeePa reduces the data transfers for synchronizing gradients and moving tensors at the cost of using fewer GPUs.

### 7.1.2 ALEXNET

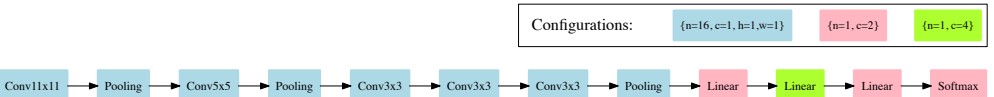

Figure 9: The global configuration for parallelizing AlexNet on 16 GPU workers.

Figure 9 shows the global configuration for AlexNet on 16 GPU workers. Note that DeePa selects the parallelism configuration that optimizes the performance for each layer. Table 2 lists the cost for different configurations of the first fully-connected layer. The standard image parallelism configuration eliminates the cost for transferring the input tensors but introduces additional data transfers for synchronizing gradients. The OWT approach completely eliminates gradient synchronization at the cost of replicating the input tensors on every GPU worker. The configuration chosen by DeePa only uses 2 GPU workers for training the first fully-connected layer, which prolongs the compute time but significantly reduces the cost for both transferring input tensors and synchronizing gradients. As a result, DeePa reduces the total cost by 5× compared to other approaches.

DeePa uses image parallelism for all convolutional and pooling layers, because the additional data transfer cost introduced by transforming configurations outweighs any performance benefits.

### 7.1.3 VGG-16

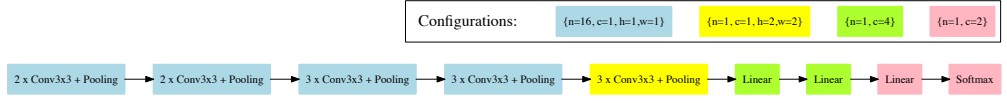

Figure 10: The global configuration for parallelizing VGG-16 on 16 GPU workers.

Table 3: The cost for different configurations for the last three convolutional layers of VGG-16.

| Configuration | # GPU Workers | Transfer Cost | Compute Cost | Update Cost | Total Cost | Parallelism Configurations |
|---|---|---|---|---|---|---|
| {n=16, c=1, h=1, w=1} | 16 | 0 | +15.9 | +134.4 | = 150.3 | Image Parallelism & OWT |
| {n=8, c=1, h=1, w=1} | 8 | 39.2 | +31.8 | +67.2 | = 138.2 | |
| {n=4, c=1, h=1, w=1} | 4 | 39.2 | +63.7 | +33.6 | = 136.5 | |
| { n=1, c=1, h=2, w=2} | 4 | 39.2 | +54.7 | +33.6 | = **127.5** | DeePa |
| {n=2, c=1, h=1, w=1} | 2 | 39.2 | +127.4 | +16.8 | = 183.4 | |

DeePa uses similar configurations for parallelizing the fully-connected layers in VGG-16 (Figure 10). In addition, DeePa also uses a different configuration to cooperatively accelerate the last three convolutional layers (the yellow node in Figure 10). Table 3 lists the cost for different parallelism configurations for the last three convolutional layers. The configuration with optimal total cost uses only four GPU workers for the last three convolutional layers to reduce data transfers for synchronizing gradients. DeePa also exploits parallelism in the height and width dimensions to further reduce the compute time.

### 7.1.4 INCEPTION-V3

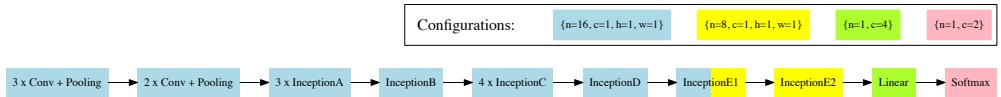

Figure 11: The global configuration for parallelizing Inception-v3 on 16 GPU workers. Each module is shown as a single node for simplicity.

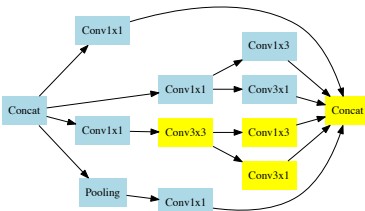

Figure 12: The configurations for parallelizing the InceptionE1 module.

The Inception-v3 model has multiple Inception *modules* (Szegedy et al., 2016). Each module has several branches of convolutional and pooling layers, which are then concatenated as the output tensor of the module. Figure 11 shows the global configuration for Inception-v3. DeePa uses different configurations to parallelize different branches for the InceptionE1 module, as shown in Figure 12. We found that this configuration reduces data transfers by 30% in InceptionE1 and InceptionE2 and reduces overall data transfers by 20%.

### 7.2 MINIBATCH SIZE

The minibatch size plays an important rule on the performance of CNNs. Figure 13 compares DeePa, PyTorch, and TensorFlow with different minibatch sizes. All three networks were trained on 16 Tesla K80 GPUs on a single node, as described in Section 7.1. We were not able to train VGG-16 and Inception-v3 with a minibatch size of 2048 images, because the required metadata size exceeds the aggregate memory capacity of the 16 GPUs.

Figure 13 shows that, DeePa achieves constant speedups compared to PyTorch and TensorFlow for various minibatch sizes. In particular, DeePa achieves 4.6-6.5×, 1.6-1.9×, and 1.2-1.5× speedup for AlexNet, VGG-16, and Inception-v3, respectively.

### 7.3 MULTI-NODE RESULTS

We evaluate the scalability of different frameworks by comparing their training throughput with different number of GPUs and compute nodes. The experiments were performed on a GPU cluster

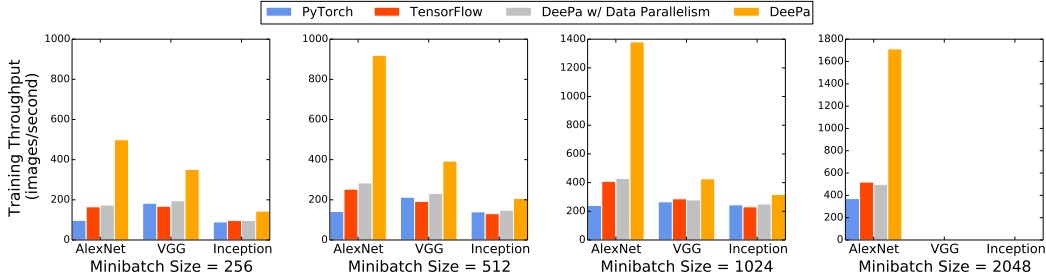

Figure 13: Performance comparisons among DeePa, PyTorch, and TensorFlow with different mini-batch sizes.

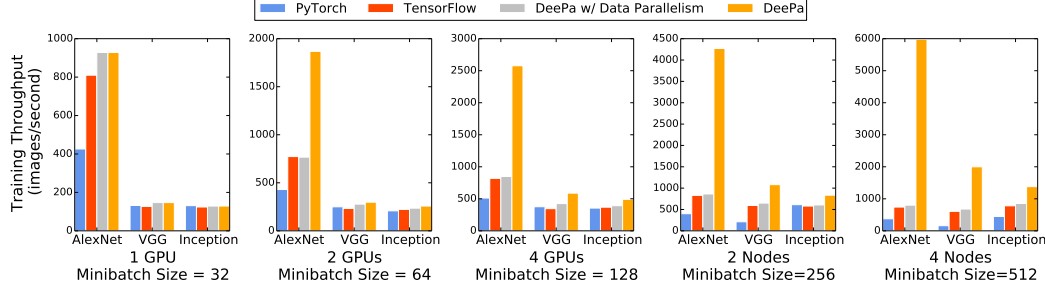

Figure 14: Performance results for DeePa, PyTorch, and TensorFlow on up to 4 nodes. We perform weak-scaling, where each GPU worker processes a batch of 32 images in every iteration. We use all 4 GPUs on each node in the last two subfigures.

with 4 nodes, each of which is equipped with two Intel 10-core E5-2600 Xeon processors, 256G main memory, and four NVIDIA Tesla P100 GPUs. GPUs on the same node are connected by NVLink, and nodes are connected over 100Gb/s EDR Infiniband.

Figure 14 shows the performance comparison among DeePa, PyTorch, and TensorFlow for weak-scaling. DeePa achieves competitive performance compared to PyTorch and TensorFlow for training on a single GPU, in which all three frameworks place all operations on a single GPU. For training on 4 GPUs on a single node, DeePa achieves $3.1\times$, $1.6\times$, and $1.3\times$ speedup for AlexNet, VGG-16, and Inception-v3, respectively. DeePa achieves even better performance speedups for trainings on multiple nodes, where the data transfer time becomes a larger component of the per-iteration training time. For training on 4 nodes, DeePa achieves $8.1\times$, $3.2\times$, and $1.8\times$ speedup for AlexNet, VGG-16, and Inception-v3, respectively.

## 8 CONCLUSION

We have presented DeePa, a deep learning framework that explores parallelism in all parallelizable dimensions to accelerate the training of CNNs. DeePa optimizes the parallelism configuration chosen at the granularity of individual layers. DeePa achieves up to $6.5\times$ for training CNNs and reduces overall data transfers by up to $23\times$ compared to state-of-the-art deep learning frameworks.

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

## A APPENDIX

### A.1 NODE AND EDGE ELIMINATION

We prove the correctness of the node and edge eliminations in Section 4. In particular, we prove that after applying node and edge eliminations, the modified graph has the same optimal configuration as the original graph.

#### A.1.1 NODE ELIMINATION

For a given computation graph $G = (V, E)$, applying a node elimination on $w$ requires $w$ having a single in-edge $e_1 = (u, w)$ and a single out-edge $e_2 = (w, v)$. The node elimination results in a modified graph $G' = (V', E')$, where $V' = V - \{w\}$, $E' = E - e_1 - e_2 + e'$, and $e' = (u, v)$.

**Theorem 1.** *Consider graphs $(V, E)$ and the result of a single node elimination $(V', E')$. Then an optimal configuration of $V, E)$ is also an optimal configuration of $(V', E')$, and an optimal configuration of $(V', E')$ is extensible to a an optimal configuration of $(V, E)$.*

*Proof.* The *Cost* function is defined in Equation 1. Let $g$ be any configuration. We first compute the difference between $Cost(g, (V, E))$ and $Cost(g, (V', E'))$.

$$
\begin{aligned}
&Cost(g, (V, E)) - Cost(g, (V', E')) \\
&= \sum_{v \in V} \{v.compute(g(v)) + v.update(g(v))\} + \sum_{e=(u,v) \in E} e.xfer(g(u), g(v)) \\
&\quad - \sum_{v \in V'} \{v.compute(g(v)) + v.update(g(v))\} + \sum_{e=(u,v) \in E'} e.xfer(g(u), g(v)) \\
&= w.compute(g(w)) + w.update(g(w)) \\
&\quad + e_1.xfer(g(u), g(w)) + e_2.xfer(g(w), g(v)) - e'.xfer(g(u), g(v))
\end{aligned}
\tag{4}
$$

Now assume $g$ is an optimal configuration for $(V, E)$. Then we have

$$
\begin{aligned}
&w.compute(g(w)) + w.update(g(w)) + e_1.xfer(g(u), g(w)) + e_2.xfer(g(w), g(v)) \\
&= \min_{c_w} \{w.compute(c_w) + w.update(c_w) + e_1.xfer(g(u), c_w) + e_2.xfer(c_w, g(v))\}
\end{aligned}
\tag{5}
$$

Therefore, $g$ is an optimal configuration of $(V', E')$. For the other direction, note that if $g$ is an optimal configuration of $(V', E')$, then it can be extended to an optimal configuration of $(V, E)$ by adding the node $w$ with the same minimal assignment. $\square$

### A.1.2 EDGE ELIMINATION

For a computation graph $G(V, E)$, applying an edge elimination on $e_1 = (u, v)$ and $e_2 = (u, v)$ results in a modified graph $G' = (V, E')$, where $E' = E - e_1 - e_2 + e'$ and $e' = (u, v)$. We prove that $Cost(g, (V, E)) = Cost(g, (V, E'))$ for any global configuration $g$ of $(V, E)$.

**Theorem 2.** *For any global configuration $g$ of graph $G = (V, E)$, $Cost(g, (V, E)) = Cost(g, (V, E'))$, where $(V, E')$ is the modified graph of $(V, E)$ after an edge elimination.*

*Proof.* We compute the difference between $Cost(g, (V, E))$ and $Cost(g, (V, E'))$.

$$
\begin{aligned}
&Cost(g, (V, E)) - Cost(g, (V, E')) \\
&= e_1.xfer(g(u), g(v)) - e_2.xfer(g(u), g(v)) + e'.xfer(g(u), g(v)) \\
&= 0
\end{aligned}
\tag{6}
$$

The last equation uses Equation 3. $\square$

### A.2 RELATED WORK ON OVERLAPPING COMMUNICATION WITH DATA TRANSFER

The overlap optimization in Section 5 is motivated by Goyal et al. (2017), which performs gradient aggregation in parallel with back propagation to scale *synchronous* training to large number of GPUs. We extend their design and implementation by also enabling the optimization for *asynchronous* training in DeePa.

### A.3 PROFILING RESULTS

We show profiling results for visualizing the performance bottlenecks in different parallelism approaches. The experiment was performed on a single node with four Tesla P100 GPUs (as described in Section 7.3). We enable overlapping computation with data transfers (described in Section 5) in this experiment.

Figure 15 shows the profiling results for training VGG-16 on 4 GPUs with different parallelism configurations. Note that DeePa with image parallelism achieves 10% higher training throughput compared to PyTorch and TensorFlow, as shown in Figure 14. Figure 15a shows that all GPUs are highly utilized during forward and backward passes, as indicated by the tight packing of tasks in the timeline. However, the image parallelism approach requires moving 4GB of metadata in every iteration, which cannot be fully overlapped with back propagation, therefore the image parallelism approach has a performance gap between iterations (shown as the white space on the GPU timelines).

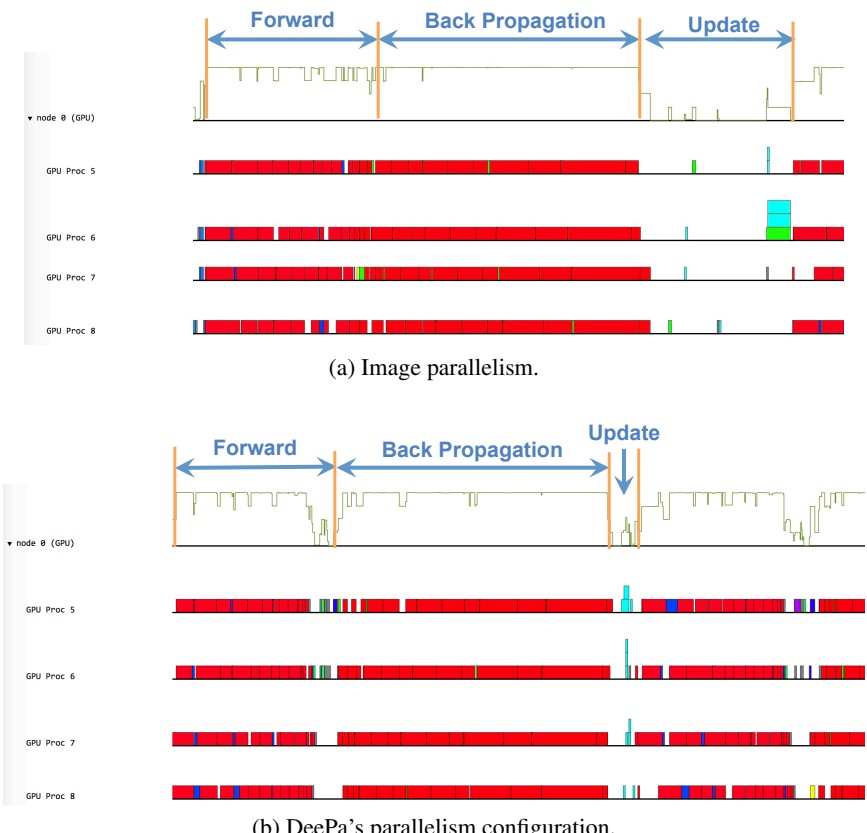

(a) Image parallelism.

(b) DeePa's parallelism configuration.

Figure 15: Timelines for training VGG-16 with a minibatch size of 128 images on 4 GPUs. The first horizontal line in each figure shows the overall GPU utilization at different timesteps, and each of the following lines shows the run times for individual operations on each GPU.

Figure 15b shows the profiling of the optimal parallelism configuration chosen by DeePa, which uses image parallelism on 4 GPUs for all convolutional layers and pooling layers and uses model parallelism on 2 GPUs for the fully connected layers. Therefore, the training with the optimal configuration includes data transfers for each fully connected layers, which adds small performance gaps at the end of the forward pass and the beginning of the backward pass (shown as the small white space on the GPU timelines). However, the optimal configuration reduces the per-iteration data transfers from 4GB to 490MB, which effectively hides data transfer overhead and achieves better GPU utilization. As a result, the optimal configuration reduces the per-iteration training time from 0.34 seconds to 0.24 seconds.

## A.4    IMANGENET-22K

We compare the performance of DeePa, PyTorch, and TensorFlow on the ImageNet-22K dataset (Russakovsky et al., 2015) that contains 21,841 different categories (the ImageNet dataset used in Section 7 contains 1,000 catagories). The last fully-connected layer in AlexNet, VGG-16, and Inception-v3 originally have 1,000 neurons followed by a 1,000-way softmax layer. To train the three networks on the ImageNet-22K dataset, we change the last fully-connected layer to have 21,841 neurons and use a 21,841-way softmax layer at the end. The modified networks were trained on 16 Tesla K80 GPUs on a single node with a minibatch size of 512 images.

Figure 16 compares the training throughput and per-iteration data transfers among DeePa, PyTorch, and TensorFlow on the ImageNet and ImageNet-22K datasets. Figure 16a shows that, on the ImageNet-22K dataset, the training throughput of PyTorch and TensorFlow is reduced by 20%-45%, while DeePa's throughput falls off by 3%, compared to training on the original ImageNet

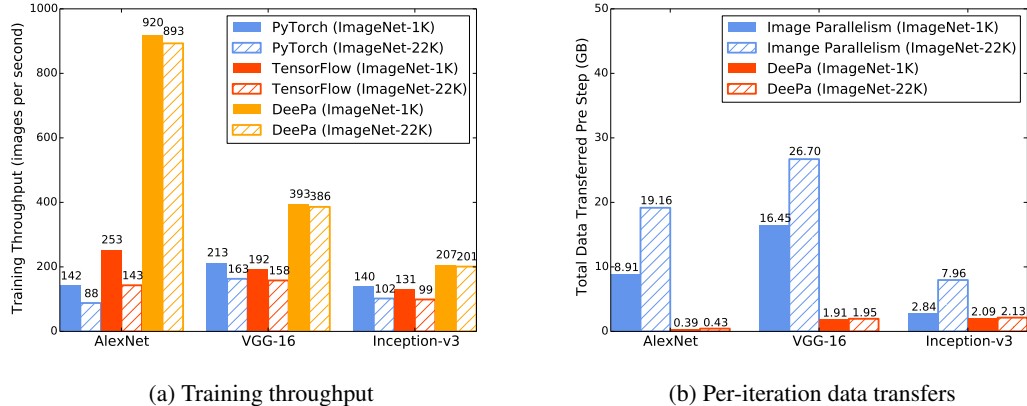

(a) Training throughput

(b) Per-iteration data transfers

Figure 16: Performance comparisons among DeePa, PyTorch, and TensorFlow on the ImageNet-22k dataset.

dataset. Figure 16b compares the per-iteration data transfers between image parallelism and the global configurations used by DeePa. Using image parallelism increases the data transfers in each iteration by 5-10GB, while DeePa only increases the per-iteration data transfers by 40MB. As a result, for training on the ImageNet-22K dataset, DeePa reduces the per-iteration data transfers by 3.7-44.5× compared to image parallelism.

