# OpenReview forum: "Exploring the Hidden Dimension in Accelerating Convolutional Neural Networks"
_ICLR.cc/2018/Conference — Reject_

### Official Review · AnonReviewer2 · 2017-11-27
**Needs more data to support**

**Rating:** 4
**Confidence:** 5

**Review:**

This paper develops a framework for parallelization of convolutional neural nets. In the framework, parallelism on different dimensions are explored for convolutional layers to accelerate the computation. An algorithm is developed to find the best global configuration.

The presentation needs to be more organized, it is not very easy to follow.

1. Computation throughput is not defined.

2. Although the author mentions DeePa with Tensorflow or Pytorch several times, I think it is not proper to make this comparison. The main idea of this paper is to optimize the parallelization scheme of CNN, which is independent of the framework used. It is more useful if the configuration searching can be developed on tensorflow / pytorch.

3. The per layer comparison is not very informative for practice because the data transfer costs of convolution layers could be completely hidden in data parallelization. In data parallelism, the GPU devices are often fully occupied during the forward pass and backward pass. Gaps are only in between forward and backward, and between iterations. Model parallelism would add gaps everywhere in each layer. This could be more detrimental when the communication is over ethernet. To be more convincing, it is better to show the profile graph of each run to show which gaps are eliminated, rather than just numbers.

4. The batch size is also a crucial factor, difference batch size would favor different methods. More comparisons are necessary.

---

> ### Author Response · Authors · 2017-12-18
> **The profiling results show that DeePa does a better job on reducing data transfers, overlapping data transfers with computation, and improving GPU utilization.**
>
> We thank the reviewer for the constructive comments.
>
> 1. Computation throughput in Figure 1 is not defined.
> We have added a definition of computation throughput in Figure 1.
>
> 2. It is more useful if DeePa can be developed on TensorFlow or PyTorch.
> We agree that our results show such implementations would be useful. Legion is the only framework that supports partitioning/parallelization across all the interesting dimensions (image, height, width, and channel for 2D CNNs), which is why we selected it to demonstrate that substantial speedups in deep learning through exploiting other parallelizable dimensions is even possible.
>
> 3. To be more convincing, it is better to show the profile graphs of different runs to help understand which gaps are eliminated.
> We have added some profiling results (Appendix A.3) to compare the performance between image parallelism and DeePa's configuration. The profiling results show that the better configurations reduce data transfers, better overlap data transfers with computation, and improve GPU utilization.
>
> 4. Performance comparisons with different batch sizes are missing.
> In the revised paper, we have added an experiment (Section 7.2) to compare the performance of the different frameworks with various minibatch sizes. The results show that DeePa achieves speedups compared to PyTorch and TensorFlow with all of the tested minibatch sizes.

---

### Official Review · AnonReviewer3 · 2017-11-29
**Potential for accelerating CNNs**

**Rating:** 5
**Confidence:** 4

**Review:**

The paper proposes an approach that offers speedup on common convolutional neural networks. It presents the approach well and shows results comparing with other popular frameworks used in the field.

Originality
- The automation of parallelism across the different dimensions in each of the layers appears somewhat new. Although parallelism across each of the individual dimensions has been explored (batch parallel is most common and best supported, height and width is discussed at least in the DistBelief paper), automatically exploring this to find the most efficient approach is new. The splitting across channels seems not to have been covered in a paper before.

Significance
- Paper shows a significant speedup over existing approaches on a single machine (16 GPUs). It is unclear how well this would translate across machines or to more devices, and also on newer devices - the experiments were all done on 16 K80s (3 generations old GPUs). While the approach is interesting, its impact also depends on the speedup on the common hardware used today.

Pros:
- Providing better parallelism opportunities for convolutional neural networks
- Simple approach to finding optimal global configurations that seems to work well
- Positive results with significant speedups across 3 different networks

Cons:
- Unclear if speedups hold on newer devices
- Useful to see how this scales across more than 1 machine
- Claim on overlapping computation with data transfer seems incorrect. I am pretty sure TensorFlow and possibly PyTorch supports this.

Questions:
- How long does finding the optimal global configuration take for each model?

---

> ### Author Response · Authors · 2017-12-18
> **Experiments with the newest generation GPUs on up to 4 machines shows that DeePa achieves even better results.**
>
> We thank the reviewer for the constructive comments.
>
> 1. It is unclear if the performance speedup still holds on multiple machines and newer GPU devices?
> In the revised paper, we have added an experiment (Section 7.3) with NVIDIA Tesla P100 GPUs (newest generation GPUs) on up to 4 compute nodes. The result shows that DeePa achieves even better performance speedups compared to PyTorch and TensorFlow for multi-node executions, where data transfer cost becomes a bigger factor in the per-iteration training time.
>
> 2. Claim on overlapping computation with data transfer seems incorrect.
> This is a fair point; it's not clear whether the current version of TensorFlow overlaps communication and computation or not (at least the version described in the original paper, Abadi et al., 2016, appears to not support such overlap, but that may have changed).  We have removed this statement from the paper. Note that all the performance comparisons are with whatever TensorFlow actually does in the version r1.3.
>
> 3. How long does finding the optimal global configuration take for each model?
> In the revised paper, we report the times for finding the optimal global configurations in the first paragraph of Section 7.1. In particular, it takes 0.7, 1.1, and 4.8 seconds for finding the optimal configurations for AlexNet, VGG-16, and Inception-v3, respectively. The reported numbers also include the time to measure the average execution time for different operations.

---

### Official Review · AnonReviewer4 · 2017-11-30
**Review of Exploring the Hidden Dimension in Accelerating Convolutional Neural Networks**

**Rating:** 7
**Confidence:** 4

**Review:**

The paper proposes a deep learning framework called DeePa that supports multiple dimensions of parallelism in computation to accelerate training of convolutional neural networks.  Whereas the majority of work on parallel or distributed deep learning partitions training over bootstrap samples of training data (called image parallelism in the paper), DeePa is able to additionally partition the operations over image height, width and channel.  This gives more options to parallelize different parts of the neural network.  For example, the best DeePa configurations studied in the paper for AlexNet, VGG-16, and Inception-v3 typically use image parallelism for the initial layers, reduce GPU utilization for the deeper layers to reduce data transfer overhead, and use model parallelism on a smaller number of GPUs for fully connected layers.  The net is that DeePa allows such configurations to be created that provide an increase in training throughput and lower data transfer in practice for training these networks.  These configurations for parellism are not easily programmed in other frameworks like TensorFlow and PyTorch.

The paper can potentially be improved in a few ways.  One is to explore more demanding training workloads that require larger-scale distribution and parallelism.  The ImageNet 22-K would be a good example and would really highlight the benefits of the DeePa in practice.  Beyond that, more complex workloads like 3D CNNs for video modeling would also provide a strong motivation for having multiple dimensions of the data for partitioning operations.

---

> ### Author Response · Authors · 2017-12-18
> **DeePa achieves even better performance speedups on ImageNet-22K**
>
> We thank the reviewer for the constructive comments.
>
> 1. The ImageNet 22-K would be a good example and really highlight the benefits of DeePa in practice.
> We have added a performance comparison (Appendix A.4) on the ImageNet-22K dataset. The results show that DeePa achieves almost the same training throughput on ImageNet and ImageNet-22K, while PyTorch and TensorFlow reduces the training throughput by 20%-45% on ImageNet-22K. In addition, the global configurations used by DeePa also reduce per-iteration data transfers by 3.7-44.5x compared to image parallelism.
>
> 2. More complex workloads like 3D CNNs for video modeling would provide a strong motivation.
> We agree that 3D CNNs would be a good example.  However, they would require significantly more engineering effort and we believe the results we have obtained (e.g., the latest ImageNet-22K numbers) already strongly support the thesis that alternative data partitioning strategies can substantially speed up deep learning.

---

### Author Response · Authors · 2018-01-11
**Revised Manuscript**

We have attempted to address the reviews' comments in the revised manuscript, and we believe the revisions have resulted in a significantly improved manuscript. The revised manuscript includes the following major changes:

1. We have added an experiment (Section 7.3) with NVIDIA Tesla P100 GPUs (newest generation GPUs) on up to 4 compute nodes to compare different frameworks for distributed executions.

2. We have conducted performance comparisons (Section 7.2) with different batch sizes and show that we are able to achieve speedups compared to PyTorch and TensorFlow with all of the tested batch sizes

3. We have added some profiling results (Appendix A.3) to visualize the performance bottlenecks in image parallelism and our configurations. The profiling results show that our configurations reduce data transfers, better overlap data transfers with computation, and improve GPU utilization.

4. We have added a performance comparison (Appendix A.4) on the ImageNet-22K dataset and show that DeePa achieves even better speedups compared to PyTorch and TensorFlow on the ImageNet-22K dataset.

---

> ### Comment · AnonReviewer3 · 2018-01-12
> **Benchmark comparisons seem inaccurate**
>
> For TensorFlow the reported numbers seem significantly lower than benchmarks at https://www.tensorflow.org/performance/benchmarks and for less optimal batch size.
>
> Can't find similar benchmarks for PyTorch so don't have a better comparison, but it opens the question of how good is that benchmark.

---

> > ### Author Response · Authors · 2018-01-12
> > **We use synchronous training for the TensorFlow experiments**
> >
> > We thank the reviewer for the constructive feedback.
> >
> > For the TensorFlow experiments, we use synchronous training with a batch size of 32 and train the models on the ImageNet dataset. The performance numbers reported on https://www.tensorflow.org/performance/benchmarks are measured by using the asynchronous training method, which has better performance than synchronous training as shown in the TensorFlow paper.
> >
> > To test the effectiveness of our TensorFlow benchmark, we rerun the TensorFlow experiments with asynchronous training and report the numbers in the following table, which shows that our TensorFlow benchmark has very similar performance compared to the benchmark on the website.
> >
> > Table 1. TensorFlow training throughput (#images/second) with a batch size of 32 on the ImageNet dataset.
> >
> > #P100 GPUs       Model                    Our numbers             Our numbers          Numbers from the website
> >                                                             Synchronous              Asynchronous         Asynchronous
> > 1                           VGG-16                 137                               141                            144
> > 2                           VGG-16                 253                               252                            253
> > 4                           VGG-16                 406                               459                            457
> > 1                           Inception3           124                               126                            130
> > 2                           Inception3           243                               255                            257
> > 4                           Inception3           439                               488                            507

---

### Decision · Program_Chairs · 2018-01-29
**ICLR 2018 Conference Acceptance Decision**

**Decision:**

Reject

**Comment:**

While this paper has some very interesting ideas the majority view of the reviewers and their aggregate numerical ratings are just too low to warrant acceptance.